# Pulmonary Rehabilitation in Patients with COVID-19—A Protocol for Systematic Review and Meta-Analysis

**DOI:** 10.3390/ijerph192113982

**Published:** 2022-10-27

**Authors:** Yanan Gao, Huiming Huang, Chunxia Ni, Yong Feng, Xiao Dong, Yin Wang, Junwu Yu

**Affiliations:** 1Faculty of Sport Science, Research Academy of Grand Health, Ningbo University, Ningbo 315211, China; 2Ningbo College of Health Sciences, Ningbo 315099, China

**Keywords:** COVID-19, pulmonary rehabilitation, systematic review, exercise capacity, pulmonary function, quality of life

## Abstract

Introduction: Pulmonary rehabilitation (PR) is a well-established treatment for patients with chronic lung disease; however, its role in patients with COVID-19 has not been systematically studied. We provide a protocol outlining the methods and analyses that will be used in the systematic review. Methods: The methodology of this systematic review protocol has been filed in PROSPERO under the registration number CRD42022301418. Five electronic databases (PubMed, Web of Science, Cochrane Library, EBSCO, and CNKI databases) will be searched from 2019 to 28 July 2022, using pre-determined search terms. Eligibility criteria will be defined using a PICOS framework. Pulmonary function, exercise capacity, and health-related quality of life will be the primary outcomes. Quantitative findings will be narratively synthesized, whilst argument synthesis combined with refutational analysis will be employed to synthesize qualitative data. Results: The results will be presented by both meta-analysis and qualitative analysis. Conclusion: This protocol describes what will be the first systematic review to conduct a worldwide assessment of the effect of PR in patients with COVID-19. Because this is a systematic review and meta-analysis, no ethical approval is needed. The systematic review and meta-analysis will be published in a peer-reviewed journal and disseminated both electronically and in print.

## 1. Introduction

The world is now in the middle of the COVID-19 pandemic. Globally, as of 2 August 2022, there have been 575,887,049 confirmed cases of COVID-19, including 6,398,412 deaths, reported to WHO [1]. COVID-19 is an acute infectious illness that primarily affects the respiratory system and lungs, with severe lung damage being the leading cause of mortality [2]. Given the commonly severe involvement of multiple organs and body functions during COVID-19, other abnormalities may persist after the acute phase has passed, potentially affecting patients’ well-being.

These issues can be addressed through pulmonary rehabilitation (PR). PR, defined as a comprehensive non-pharmacological strategy, is often remarkably successful in patients with chronic obstructive pulmonary disease (COPD) [3]. However, COVID-19 is distinct from other types of lung disease, and understanding the impact of PR on COVID-19 patients is critical to developing better COVID-19 treatments.

According to the 2013 American Thoracic Society (ATS)/European Respiratory Society (ERS) Statement [3], PR is “a comprehensive intervention based on a comprehensive patient assessment and patient-tailored treatment, including but not limited to exercise training, education, and behavioral changes. PR aims at improving chronic respiratory, physical, and mental status of patients with disease, and promoting long-term adherence to health care behavior”. Since 2015, numerous clinical trials have provided data on safety and clinical outcomes of planned PR models, including home rehabilitation [4]; telerehabilitation [5]; interactive, web-based models [6]; mixed heart failure/PR models [7]; and so on [8]. The comprehensive assessment of PR mainly includes five areas: exercise capacity, quality of life, dyspnea, nutritional status, and occupational status [9].

PR is the cornerstone of the treatment of patients with chronic respiratory diseases [10]. There is direct evidence that PR can improve exercise capacity, reduce dyspnea, improve health-related quality of life, and reduce hospitalization rates in chronic obstructive pulmonary disease (COPD) patients [11,12]. There is now growing evidence that PR can improve outcomes in other conditions such as interstitial lung disease [13], bronchiectasis [14], and pulmonary hypertension [15]. Furthermore, education and psychological support in PR can enhance patients’ cognition and comprehension of their diseases [16]. Therefore, the extension of PR has great practical significance, and many nations or territories have also issued guidelines and consensus statements for PR nursing in chronic respiratory diseases [17,18,19,20].

In recent years, there have been several works discussing the effects of PR on COVID-19 patients. Gonzalez-Gerez et al. [21] found that PR can improve the physical condition, dyspnea, and perceived effort among people with mild to moderate COVID-19 symptoms in the acute stage; however, long-term effects cannot be determined based on the results in the study. A retrospective cohort study [22] found that PR could be used to promote exercise capacity improvement after COVID-19. Their study, however, can only suggest a potential association between PR and outcomes, not the causal effect of PR, and is highly biased. Conducting high-quality clinical trials in a large number of patients is nearly impossible due to the high contagiousness of COVID-19, and a systematic review and meta-analysis are still lacking. It is still unclear how PR affects COVID-19 patients and whether this impact is affected by the patient’s age, disease severity, stage of disease, type of PR, and the program of PR.

A synthesis of the evidence of the association between PR and pulmonary function, exercise capacity, and health-related quality of life would contribute to a better understanding of the relationship between PR and patients with COVID-19. Furthermore, investigating the prognostic value of PR for the above outcomes would aid in better understanding the mechanism of COVID-19 and making better clinical decisions. Unfortunately, most of the current PR recommendations for COVID-19 rely on previous inferences from severe acute respiratory syndrome (SARS) recovery [23].

Stronger conclusions can be drawn from a systematic review of the literature than from any single study, and this protocol will outline the methods and analyses used in a systematic review. The systematic review and meta-analysis will explore whether PR is an effective intervention to improve the prognosis of patients with COVID-19.

## 2. Materials and Methods

### 2.1. Aims

#### 2.1.1. Primary Aim

The aim of this systematic review and meta-analysis protocol is to evaluate the effect of pulmonary rehabilitation (PR) on COVID-19 patients.

#### 2.1.2. Review Questions

What are the effects of PR on COVID-19 patients compared to controls?

Which outcomes are significantly influenced by using PR?

Whether the effect of PR intervention is different on COVID-19 patients with different disease severity and intervention frequency?

Which types and characteristics of interventions with PR were devised for COVID-19 patients?

Whether PR is an effective intervention for improving the prognosis of patients with COVID-19?

### 2.2. Design

This review is conducted in accordance with the Preferred Reporting Items for Systematic Reviews and Meta-Analyses (PRISMA 2020 [24]) statement. The systematic review protocol was prospectively registered at the International Prospective Register of Systematic Reviews (PROSPERO, Registration No. CRD42022301418). The intent of the systematic review, as registered in PROSPERO, was to evaluate the effects of PR in patients with COVID-19, and to explore whether PR is an effective intervention for improving the prognosis of patients with COVID-19. If the conditions for effect size merging are met, meta-analysis will exist.

#### 2.2.1. Inclusion and Exclusion Criteria

The criteria of this study will be summarized based on the participants, interventions, comparisons, outcomes, and study design (PICOS) schema according to the Cochrane handbook for systematic reviews of interventions.

##### Types of Studies

Include: All empirical research studies published in peer-reviewed journals that provide comparative quantitative data on our primary outcomes will be considered as eligible, such as randomized controlled trials (RCTs), quasi-RCTs, non-randomized controlled trials, and observational case-control and cohort studies. For relevant but unpublished studies, research groups will be contacted with a request to provide summary data.

Exclude: We will exclude reviews, letters, poster presentations, editorials, case series, and protocols.

##### Types of Participants

Include: All patients suffering from COVID-19 will be included regardless of stage of disease, severity of illness, sex, age, race, education, and economic status. The diagnostic criteria refer to clinical diagnosis and treatment guidelines issued by the United States [25] and China [26].

##### Types of Interventions

Include: A broad range of PR will be included to gain a comprehensive overview of current approaches to PR. Interventions will be included if they match the following definition of PR: “the delivery of rehabilitation services which primarily takes advantage of respiratory training, exercise training, education, and behavioral changes”. Respiratory rehabilitation-based exercise training interventions are also incorporated, such as respiratory muscle training, diaphragm training, and traditional Chinese exercise training. Telerehabilitation, face-to-face PR, in-person PR, supervised PR, and unsupervised PR will all be included in the study.

Exclude: Respiratory rehabilitation with the assistance of a respiratory trainer. Exercise training alone that is not based on respiratory rehabilitation, such as endurance and aerobic training.

##### Types of Outcomes

Include: Outcomes that may be included are listed in Table 1.

Exclude: Studies that do not contain data on either of the above outcomes.

### 2.3. Search Methods

#### 2.3.1. Search Strategy

##### Electronic Searches

The review will involve searching the PubMed, Web of Science, Cochrane Library, EBSCO, and CNKI databases from December 2019 to July 2022. Medical subject heading (MeSH) terms will be adopted to search the database, mainly including ”COVID-19” AND ”Rehabilitation” AND ”Pulmonary” AND ”Trial”. Each database will use subject words and free words to search. All searches will be limited to the English and Chinese languages, but no geographical restrictions will be applied. Table 2 lists the search strategies and search words of the PubMed database in detail. The search strategies of other databases will convert the logical operators and search fields accordingly. Different search strategies will be used for different language databases. PubMed is used as an example, and the specific search strategy is detailed in Table 2.

##### Additional Resources

We will also manually search the following resources to identify ongoing or completed clinical trials, such as Google Scholar (http://scholar.google.com), Baidu Scholar (http://xueshu.baidu.com/), Clinical Trials (http://www.clinicaltrails.gov), and the China Clinical Trials Registry (http://www.chictr.org/cn/).

#### 2.3.2. Study Selection

EndNote (Version X9, Clarivate, Philadelphia, PA, USA) will be used for import, grouping, deduplication, adding full text, and so on. After importing the reference into EndNote, we will first filter out the literature by comparing the title, author, year, journal name, volume, page number, and other information to remove duplicate references. Two authors (Y.G., C.N.) trained in evidence-based medicine will conduct a one-by-one review of the titles and summaries of the reference books according to the inclusion and exclusion criteria and remove the references that obviously do not meet the inclusion criteria into the exclusion folder. They will add the full text of the bibliographies, meeting the requirements of the preliminary hearing. The two authors (Y.G., Y.F.) will read the contents of the research design one by one in the full text of the literature, remove the literature that does not meet the requirements to the exclusion folder, and record the reasons for exclusion in Excel. In the screening process, the solution should be discussed first if there is any disagreement. If disagreements still exist, the author (H.H.) will assist in judgment. The literature screening process is shown in Figure 1.

#### 2.3.3. Data Extraction

Data will be extracted by two independent reviewers (YG and HH) according to an agreed data extraction form (Appendix A, Appendix B, Appendix C, Appendix D and Appendix E). Any disagreements will be resolved by consensus.

Data to be extracted mainly include bibliographic information (e.g., author, title, year, publication), demographics (e.g., sex, age, sample size), groups (e.g., group name, group description, intervention frequency, intensity, duration, co-interventions), and outcomes (e.g., time points measured/reported, definition, unit of measurement, imputation of missing data). The specific data to be extracted are shown in Table 3.

Authors will be contacted if data are missing or unclear in the selection of articles. If sufficient information cannot be obtained in this way, we will analyze the available data and the potential impact of insufficient data on the study results in the discussion.

### 2.4. Quality Appraisal

The identified trials will be assessed independently by two reviewers (Y.G. and C.N.). For case-control and cohort studies, the risk of bias will be assessed using the Newcastle–Ottawa Scale (NOS) [27]. Stars are awarded for each domain, which allows the study to be graded into poor, fair, or good quality. For randomized controlled trials, quality assessment will be carried out with the Cochrane Risk of Bias Tool [28]. Any disagreements will be reviewed by the third reviewer (Y.F.) and resolved by discussion among all reviewers. If the information about the risk of bias in the clinical trial is unclear, we will try to contact the author by email.

### 2.5. Data Synthesis

It is anticipated that the included studies will vary significantly in type and method, though meta-analyses will be conducted if data are available or situations allowed. According to the heterogeneity between studies, the method of data analysis and synthesis will be determined. When we find obvious heterogeneity in the combined data, we will use subgroup analysis, sensitivity analysis, and publication bias to investigate the source of the heterogeneity.

#### 2.5.1. Meta-Analysis

If the heterogeneity between the included studies is within the acceptable range, a meta-analysis of the study results will be conducted. A Chi-square test (χ^2^) and I2 will be used to analyze the heterogeneity between the clinical trials. If *p* > 0.1, I2 ≤ 50%, it indicates that the heterogeneity between the clinical trials is within the acceptable range, and a fixed-effect model will be used to analyze the data. If *p* ≤ 0.1, I2 > 50%, indicating that the heterogeneity between clinical trials is considerable, subgroup analysis will be needed to identify the source of heterogeneity, and the random-effect model will be used to analyze the data. RevMan5.3 software (Cochrane Collaboration, Oxford, UK) will be used to synthesize the study data. Mean difference (MD) or standardized mean difference (SMD) and 95% confidence interval (CI) are used to describe the effect size of continuous data (e.g., spirometry/mL, six-minute walk/min, VO2max/(mL/min/kg), strength/kg, blood pressure/kPa, questionnaire scale scores). The Z test judges the effect size, and it has statistical significance when *p* ≤ 0.05. The data synthesis results will be presented in the form of forest plots.

#### 2.5.2. Descriptive Review

If the heterogeneity between the included studies is significant, we will make a descriptive analysis of the study results. A Chi-square test (χ^2^) and I2 will be used to analyze the heterogeneity between the clinical trials. If *p* ≤ 0.1, I2 > 75%, it indicates that the heterogeneity between the clinical trials is very significant, if the heterogeneity is substantial, we will make a narrative qualitative summary. Study comparisons will be grouped (e.g., severity of disease, sex, age, types of intervention) to answer the research questions and findings will be synthesized based on outcomes. The characteristics of included studies will be presented in a narrative format, as recommended by PRISMA.

#### 2.5.3. Subgroup Analysis

If there is a certain degree of heterogeneity between included clinical trials, subgroup analysis can be used to determine the source of the heterogeneity. The subgroup analysis will be conducted according to age, sex, country, severity of the disease, intervention length, different outcome measurement time points, and different follow-up time points.

#### 2.5.4. Sensitivity Analysis

The purpose of sensitivity analysis is to evaluate the bias variables by eliminating each study one at a time. It will compute the sensitivity of each study over the whole project to determine whether an individual study has a substantial influence on the outcomes.

### 2.6. Assessment of Publication Biases

We will assess publication bias by funnel plots for asymmetry when at least 10 trials are available [29]. If the plot is asymmetric and there is no inverted funnel form, there may be publication bias. The causes might be connected to the small sample size, allocation concealment, and insufficient blind method implementation.

### 2.7. Ethical Considerations

As this study is only a systematic review and does not involve human or animal experimentation or personal privacy, ethical approval is not required.

## 3. Results

The results of the systematic review will be published as a peer-reviewed article.

## 4. Discussion

Among patients with pulmonary diseases, rehabilitation helps reduce dyspnea, increase exercise capacity, and improve health-related quality of life [3]. Therefore, rehabilitation might be a valuable treatment in patients with COVID-19. Patients with COVID-19 often have pathological features, such as pulmonary interstitial or alveolar edema and pulmonary inflammatory lymphoid infiltration, and are prone to acute respiratory distress syndrome (ARDS), causing lung injury [30]. Patients with severe COVID-19 may experience significant decreases in lung function, potentially requiring mechanical ventilation [31]. Respiratory and circulatory failure are common causes of death among COVID-19 patients [32]. Although the mechanisms of COVID-19-induced lung injury are still being elucidated [33], pulmonary rehabilitation (PR) is necessary at any stage in the course of the COVID-19.

Integrated into the individualized treatment of the patient, PR is designed to reduce symptoms, optimize functional status, increase participation, and reduce healthcare costs through stabilizing or reversing systemic manifestations of the disease [34]. PR has emerged as a cost-effective intervention for managing chronic lung disease [35]. As evidence has shown, PR improves the 6 min walking distance (6MWD), QoL, and respiratory symptoms in patients with chronic obstructive pulmonary disease (COPD) and interstitial lung disease (ILD) [36,37]. However, not all patients with pulmonary disease benefit from PR to the same degree. In both COPD and ILD patients, studies have shown that PR is not responsive to disease rehabilitation [38,39].

Currently, studies focusing on PR in patients with COVID-19 are few. Many of the difficult questions about PR have not been answered, such as whether PR is appropriate for all COVID-19 patients, at which stage in the course of COVID-19 should PR be administered, how specifically does PR work against COVID-19 compared to other types of pneumonia, and so on. Although there are some data showing the impressive benefits of PR participation, the effectiveness of rehabilitation has not been systematically summarized yet.

Some international rehabilitation associations provided PR guidelines for COVID-19 patients. For example, the Chinese Association of Rehabilitation Medicine has established different PR programs for patients with light, moderate, severe, and worse symptoms during hospitalization and discharge [40]. The PR guideline from Turkey emphasizes that exercise training is the most effective and compulsory method in achieving the goals of PR [23]. Carda et al. [41] made suggestions for the PR program according to their clinical experience of COVID-19, and they strongly advised the implementation of teleconsultation and telerehabilitation devices and suggested that patients who had kept negative results of COVID-19 for more than 7 days after their first diagnosis could be given access to PR. The 2020 British Thoracic Society [42] also updated the PR guideline for COVID-19 from fatigue, mood disturbances, cognitive function, and support to reopening work. However, most of the current PR guidelines adopted for patients with COVID-19 are based on the experience gained during the response to the SARS epidemic in 2004. Therefore, more evidence is needed to demonstrate the impact of PR and different PR designs on COVID-19 patients. This protocol will represent the first systematic review and meta-analysis on effects of PR in patients with COVID-19. A deep understanding of PR can alleviate the COVID-19 crisis and optimize COVID-19 resource allocation. Therefore, the conclusions of the systematic review will have direct practical implications and clinical relevance.

## 5. Limitations

There will be some limitations to this systematic review. First, the outbreak of COVID-19 is sudden. It is impossible to formulate and implement a large-sample randomized controlled trial in a short period, and the quality of clinical trials may not be high enough, which affects the quality of evidence to a certain extent. Secondly, the different types, frequency, intensity, and duration of PR may cause clinical heterogeneity. Thirdly, this review will only include Chinese and English studies from the literature, which may lead to selective bias. Although there are some limitations, the team members will still carry out this review to provide some references and suggestions for clinical decision-making and further clinical research.

## 6. Conclusions

Pulmonary rehabilitation (PR) is an effective tool of rehabilitation interventions for COVID-19 patients. The findings of this systematic review and meta-analysis can help physical therapists and the general public in actively addressing the challenges posed by COVID-19.

## Figures and Tables

**Figure 1 ijerph-19-13982-f001:**
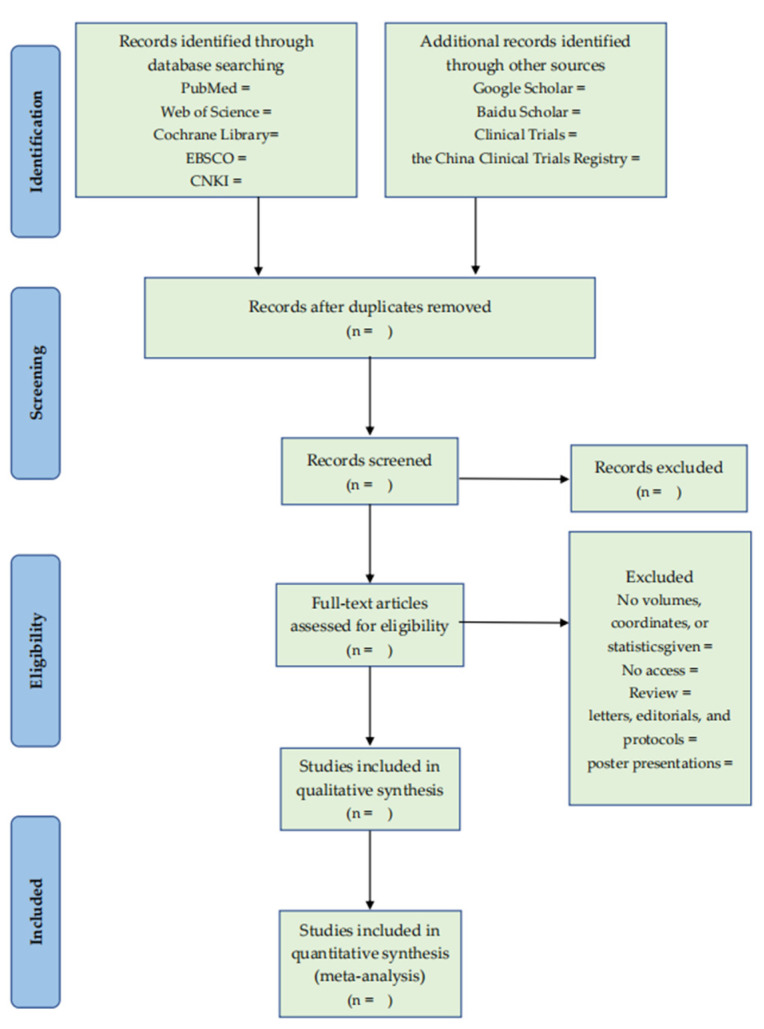
The PRISMA flow diagram of literature screening process.

**Table 1 ijerph-19-13982-t001:** Outcomes and Measurements Will be Included in the Review.

Outcomes	Assessments and Indexes
**Primary outcomes**	
Pulmonary function	Pulmonary function tests (FVC, FEV1, PEF, VC)
**Exercise capacity**	Six-minute walk test, cardiopulmonary exercise testing (VO2max, peak VO2)
Dyspnea	Multidimensional dyspnea -12 scale, Borg scale
**Health-related quality of life**	Quality-of-life inventory
**Secondary outcomes**	
**The number of hospitalizations and days in the hospital**	Admission notes, progress notes
Anxiety and depression	The hospital anxiety and depression scale
**Strength and endurance**	Muscle strength and endurance tests (HGS, 1RM, sit-ups)
Sleep disorders	Pittsburgh sleep quality index
**Fatigue**	Multidimensional fatigue inventory
Anorexia	Anorexia inventory
**Blood pressure**	Sphygmomanometer
Heart rate	Heart rate monitor
**Body composition**	Body composition analyzer (BMI, MM, LBM)
Hematological and biochemical parameters	Blood biochemical automatic analyzer (SPOZ)
**CT imaging**	CT scanner

Abbreviation: FVC, forced vital capacity; FEV1, 1 s forced expiratory volume; PEF, peak expiratory flow; VC, vital capacity; VO2max, maximal oxygen uptake; peak VO2, peak oxygen uptake; HGS, handgrip strength; 1RM, one-repetition maximum; BMI, body mass index; MM, muscle mass; LBM, lean body mass; SPOZ, oxygen content of blood.

**Table 2 ijerph-19-13982-t002:** Search Strategy for PubMed.

Number	Search Terms of Query
#1	COVID-19[Mesh] OR COVID-2019[tiab] OR COVID19[tiab] OR COVID-19 pandemic[tiab] OR COVID-19 virus disease[tiab] OR SARS-CoV-2 infection[tiab] OR 2019 novel coronavirus infection[tiab] OR coronavirus disease 2019[tiab] OR 2019-nCoV infection[tiab] OR COVID-19 virus infection[tiab]
#2	severe acute respiratory syndrome coronavirus 2[Mesh] OR SARS-CoV-2[tiab] OR 2019-nCoV[tiab] OR 2019 novel coronavirus[tiab] OR COVID-19 virus[tiab] OR COVID19 virus[tiab] OR coronavirus disease 2019 virus[tiab] OR Corona Virus Disease 2019[tiab]
#3	#1 OR #2
#4	Pulmonary[Mesh] OR pulmonary rehabilitation[tiab] OR lung rehabilitation[tiab] OR pulmonary rehabilitation exercise[tiab] OR pulmonary recovery[tiab] OR lung recovery[tiab] OR lung rehabilitation exercise[tiab] OR pulmonary rehabilitation therapy[tiab]
#5	Respiratory therapy[Mesh] OR pulmonary respiratory therapy[tiab] OR breathing therapy[tiab] OR respiratory muscle training[tiab] OR respiratory muscle exercise[tiab] OR inspiratory muscle training[tiab] OR lung respiratory therapy[tiab]
#6	#4 OR #5
#7	#3 AND #6
#8	randomized controlled trial[pt] OR controlled clinical trial[pt] OR randomized[tiab] OR clinical trials as topic[Mesh] OR trial[tiab]
#9	#7 AND #8

**Table 3 ijerph-19-13982-t003:** Variables to be extracted at the full-text stage.

**Bibliographic**	Authors	Reported conflict of interest
	Title	Study design
	Year of publication	Country
	Journal	Setting(s)
	Sources of funding	Type of allocation sequence
	Institutions and affiliations	Inclusion and exclusion criteria
**Demographics**	Age	Severity of COVID-19
	Sex	Stage of disease
	Sample size	With or without complications
		Description of health status
		Physical activity levels
**Pulmonary rehabilitation protocol**	Pulmonary rehabilitation modalities (e.g., telerehabilitation, face to face, self-management)	Rehabilitation intensity
	Pulmonary rehabilitation program (e.g., aerobic exercise, resistance exercise, nutrition, education)	Rehabilitation session volume (total time of work and recovery intervals)
	Duration, intensity, and modality	
	Exercise modality (e.g., treadmill, stationary cycle)	Pulmonary rehabilitation combined with other interventions (e.g., combination of group psychological intervention and pulmonary rehabilitation)
	Number and duration of work/recovery intervals	
	Control intervention	
**Pulmonary function**	Time points measured	Pulmonary function levels at baseline and after pulmonary rehabilitation
	Time points reported	Pulmonary function levels at baseline and after follow-up
	Follow-up time	
	Measurement method	
	Unit of measurement	
**Exercise capacity**	Time points measured	Exercise capacity at baseline and after pulmonary rehabilitation
	Time points reported	Exercise capacity at baseline and after follow-up
	Follow-up time	
	Measurement method	
	Unit of measurement	
**Dyspnea**	Time points measured	Dyspnea levels at baseline and after pulmonary rehabilitation
	Time points reported	Dyspnea levels at baseline and after follow-up
	Follow-up time	
	Types of questionnaires	
	Questionnaire scoring rules	
**Health-related quality of life**	Time points measured	Quality-of life-levels at baseline and after pulmonary rehabilitation
	Time points reported	Quality-of-life levels at baseline and after follow-up
	Follow-up time	
	Types of questionnaires	
	Questionnaire scoring rules	
**Other variables**	Days in the hospital	Adverse effects (e.g., reasons for dropout, dizziness)
	Strength and endurance	Recruitment, retention, adherence, outcome rates, and acceptability of intervention
	Sleep disorders	
	Fatigue	
	Blood pressure	
	Body composition	
	Hematological and biochemical parameters	

## Data Availability

Data sharing is not applicable to this article as no new data were created or analyzed in this study protocol.

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
