# Peer review of "Pulmonary Rehabilitation in Patients with COVID-19—A Protocol for Systematic Review and Meta-Analysis"

_ijerph, 2022, doi:10.3390/ijerph192113982_

Round 1
Reviewer 1 Report (Previous Reviewer 3)
See final comments on the attached file. Analyse and correct the present minor sugestions and this not necessary another review.

Author Response
Thank you for reviewing the manuscript, your comments are very important to us. We have adopted all the reviewer' suggestions and revised all the issues raised by the reviewer.
Response:
We have emphasized the "protocol" and modified the expression in several places in the manuscript.
Revisions:
Line 12: We provide a protocol outlining the methods and analyses that will be used in the systematic review.
Line 20: This protocol describes what will be the first systematic review to conduct a worldwide assessment of the effect of PR in patients with COVID-19.
Line 81: this protocol will outline the methods and analyses used in a systematic review.
Line 87: The aim of this systematic review and meta-analysis protocol is to evaluate the effect of pulmonary rehabilitation (PR) on COVID-19 patients.
Line 296: This protocol will represent the first systematic review and meta-analysis on effects of PR in patients with COVID-19.
Response:
We have corrected the grammatical errors in the manuscript.
Revisions:
Line 149: “data-bases” has been modified to “databases”
Line 166: “im-porting” has been modified to “importing”
Line 174: “re-move” has been modified to “remove”
Line 216: “be-tween” has been modified to “between”
Line 252: “this” has been modified to “the”
Response:
We have modified the format of the references.
Revisions:
[39] A.S. Scott, M.A. Baltzan, J. Fox, N. Wolkove, Success in pulmonary rehabilitation in patients with chronic obstructive pulmonary disease, Canadian respiratory journal 17(5) (2010) 219-223.

Reviewer 2 Report (New Reviewer)
The manuscript is A Protocol for a Systematic Review and Meta-Analysis of pulmonary rehabilitation in COVID-19 patients. The topic of the manuscript is interesting
The introduction is too long to put in situation the problem to be analyzed.
The design and methods are appropriate for the research question
The authors make a discussion about statements that they write as “PR is designed to reduce symptoms, optimize functional status, increase participation, and reduce healthcare costs through stabilizing or reversing systemic manifestations of the disease”, without our knowledge of the results of the analysis of this systematic review and meta-analisis. I think that the article must included the results of the systematic review and meta-analisis
Author Response
Thank you for reviewing the manuscript, your comments are very important to us. We have adopted all the reviewer' suggestions and revised all the issues raised by the reviewer.

Round 2
Reviewer 2 Report (New Reviewer)
ok with the corrections performed
This manuscript is a resubmission of an earlier submission. The following is a list of the peer review reports and author responses from that submission.
Round 1
Reviewer 1 Report
Dear Sirs,
The topic is interesting but more studies are needed to be able to carry out a systematic review and meta-analysis in order to reach firmer conclusions (at least studies published in the last five years), I understand that the pandemic began at the end of November but it is necessary to carry out of more studies on this subject to be able to carry out a systematic review and a meta-analysis.
In fact, the table lacks information on the number of studies found, as well as the steps followed to perform the meta-analysis. The discussion, limitations and conclusions should be improved, expanding the content related to the systematic review.
Reviewer 2 Report
The effects of pulmonary rehabilitation in COVID-19 patients – a protocol for systematic review and meta-analysis
The authors propose the method for a systematic review and meta-analysis to evaluate the effect of pulmonary rehabilitation on COVID-19 patients. The topic is very important, as robust conclusions about interventions to help patients cope with the consequences of COVID-19 are still lacking. The proposed manuscript requires revisions for English, as its present form makes it difficult to understand properly.
Also, several questions remain to be answered:
Questions:
- The period of the review / manuscript scopes from 2019 to Feb. 28th Could the authors extend this period to April/May 2022, as the literature is growing rapidly?
Aims and methods:
- “COVID-19 patients” is a vast target:
- Why not focus on patients with long-COVID?
- Or on patients experiencing dyspnea after a certain duration following infection?
- 1.2. Review questions
- Should the authors add question about duration of intervention/frequency? Dose / response?
- This point is mentioned later in the manuscript but would be relevant in the review questions.
- Outcomes:
- I would like to know if the authors will show any interest in cognitive outcomes or brain fog? These complaints are very common in long-COVID patients or patients having been hospitalized following COVID-19 infection. Please comment.
- Limitations:
- Thirdly, this review will only include in Chinese and English studies literature, which may lead to selective bias. However, it is mentioned in methods section that only English papers will be considered. Please clarify accordingly.
As mentioned, the manuscript needs to be revised for English. But here are a few English/spelling revisions:
Introduction:
- Line 33: change during January 13th for On January 13th.
- Line 43: patients with heavy or light symptoms
- Line 45: Please review this sentence: As a result, the rehabilitation of pulmonary is very critical for patients. It doesn’t make sense.
- Line 46: And pulmonary rehabilitation (PR) can address these issues. Please revise the English structure of this sentence.
Section 2. Background
- Lines 84-85 do not make sense
- The PR guideline from Turkey emphasizes that exercise training is the most effective and compulsory method in achieving the goals of PR [29].
- Please harmonize spelling: e.g. dyspnea and dyspnea
- Lines 100-101 : The meaning of the following sentence is unclear:
Due to the rapid spread of COVID-19, it is almost impossible to conduct high-quality clinical trials in a large number of patients, systematic review and meta-analysis have not yet been reported.
Reviewer 3 Report
This manuscript "The effects of pulmonary rehabilitation on COVID-19 patients—a protocol for systematic review and meta-analysis" in generally, presents an area of current interest but it does not present the results it proposes. Not having been able to carry out a systematic review due to scarce data on PR in patients with COVID-10, the authors should have analyzed the variables of the different studies grouped by similarities in order to formulate proposals for clinical practice and further research.
The english writing needs to be improved in minor parts with examples below.
1. Introduction
line 45- As a result, the pulmonary rehabilitation is very critical for patients and can address these issues (describe in what way PR is critical and clarify what issues ?)
line 48-"is often extremely successful [9]." (refer to more recent studies and guidelines/protocols on the effectiveness of RP).
2. Background
The manuscript presents the current literature regarding the different effects of PR in people with chronic respiratory disease in general and patients with COVID-19 in more recent years.
3. The review
We cannot see the sample in the PRISMA flow diagram of literature screening process.
The methods were well described.
4. Discussion
This part is not well presented because it admites that currently, studies focusing on PR in patients with COVID-19 are few and many of the difficult questions about PR have not been answered, such as whether PR is appropriate for all COVID-19 patients, which stage in the course of COVID-19 should PR be administered, and how specifically does PR work against COVID-19 compared to other types of pneumonia, and so on. Although there are some data showing the impressive benefits of PR participation, the effectiveness of rehabilitation has not been systematically summarized yet. So we can conclude that there isn´t objetive data for answer research questions.
We can affirm that is a lack of descriptive results in the selected literature in order to answer the research questions in relation to COVID-19 patients. If data are scarce, authors should have opted for other study design than a systematic review or meta-analysis, such as a narrative literature review for example.
5. Limitations and Conclusions
Authors said that although there are some limitations, the team members still carry out this review to provide some references and suggestions for clinical decision-making and further clinical research but we cannot found which are the guidelines that result from this study for clinical practice regarding the intervention of PR.